# Conductive porous vanadium nitride/graphene composite as chemical anchor of polysulfides for lithium-sulfur batteries

Zhenhua Sun[1], Jingqi Zhang[1], Lichang Yin[1], Guangjian Hu[1], Ruopian Fang[1], Hui-Ming Cheng[1,2] & Feng Li[1]

Although the rechargeable lithium–sulfur battery is an advanced energy storage system, its practical implementation has been impeded by many issues, in particular the shuttle effect causing rapid capacity fade and low Coulombic efficiency. Herein, we report a conductive porous vanadium nitride nanoribbon/graphene composite accommodating the catholyte as the cathode of a lithium–sulfur battery. The vanadium nitride/graphene composite provides strong anchoring for polysulfides and fast polysulfide conversion. The anchoring effect of vanadium nitride is confirmed by experimental and theoretical results. Owing to the high conductivity of vanadium nitride, the composite cathode exhibits lower polarization and faster redox reaction kinetics than a reduced graphene oxide cathode, showing good rate and cycling performances. The initial capacity reaches $1,471 \, mAh \, g^{-1}$ and the capacity after 100 cycles is $1,252 \, mAh \, g^{-1}$ at 0.2 C, a loss of only 15%, offering a potential for use in high energy lithium–sulfur batteries.

[1] Shenyang National Laboratory for Materials Science, Institute of Metal Research, Chinese Academy of Sciences, Shenyang 110016, China. [2] Tsinghua-Berkeley Shenzhen Institute, Tsinghua University, Shenzhen 518055, China. Correspondence and requests for materials should be addressed to F.L. (email: fli@imr.ac.cn).

arge-scale electrical energy storage involves transportation and stationary applications ranging from plug-in hybrid electric vehicles and full electric vehicles to the widespread use of intermittent renewable energy in the modern electrical grid, all of which require advanced battery systems[1]. The high capacity and low cost of lithium–sulfur (Li–S) batteries are essential for achieving practical applications[2,3]. These batteries possess high specific energy of 2,500 Wh kg$^{-1}$ and 2,800 Wh l$^{-1}$, and although their average working voltage is as low as 2.15 V, their high theoretical specific capacity of 1,672 mAh g$^{-1}$ can compensate for this limitation[4]. The practical energy density for packaged Li–S batteries may reach as high as 500–600 Wh kg$^{-1}$ or 500–600 Wh l$^{-1}$, which is sufficient for driving an electric vehicle 500 km[5–7].

Despite these attractive properties, one of the major issues with Li–S batteries is their sluggish reaction kinetics stemming from the high electronic resistivity of sulfur and lithium sulfides. As the resistivity of sulfur is as high as $10^{24} \Omega$ cm, it is necessary to be combined with conductive materials[8]. In addition, the resistivity of Li$_2$S is $>10^{14} \Omega$ cm and the Li ion diffusivity in Li$_2$S is low[9]. Once an insoluble insulation layer composed of Li$_2$S$_2$ and/or Li$_2$S is plated on the electrode, it would increase the internal resistance, resulting in polarization that decreases energy efficiency. Moreover, the 79% volume expansion of sulfur upon cycling induces the pulverization of active materials, which often results in poor contact with current collectors to further slow reaction kinetics[10]. The other major issue is polysulfides (Li$_2$S$_4$–Li$_2$S$_8$) dissolving in the electrolyte and migrating between the anode and the cathode, which causes the so-called 'shuttle effect' in a process in which polysulfides participate in reduction reactions with lithium and re-oxidation reactions at the cathode[11,12]. Despite the fact that the shuttle effect provides an

overcharge protection, it causes low discharge energy capacity, thermal effects, self-discharge and low Coulombic efficiency[13,14].

Porous carbon-based materials used as barriers and hosts have been demonstrated to be a simple approach to suppress the polysulfide shuttle effect[15–18]. Owing to the large specific surface area, macropores and mesopores can encapsulate a large amount of sulfur and facilitate fast ion transport[19]. A microporous sulfur/carbon composite has been produced that had an unusual capacity between 1.5 and 2 V, indicating a mixture of the two elements at the atomic level[20]. Nevertheless, because of the distinct non-polarity of carbon and the polarity of the Li$_2$S$_n$ species, the confinement of polysulfides inside the pores is mainly a result of weak physical interactions[21]. Some advantages of porous carbon are conflicting; for instance, a large surface area of Li$_2$S$_2$ and Li$_2$S deposition is prone to cause an open structure and lead to ineffective trapping of polysulfides[22], but a small pore volume limits the sulfur loading[23,24]. Functionalized graphene materials, such as graphene oxide obtained by the hydrothermal method, are decorated with hydroxyl and epoxide functional groups, and have chemical interactions with polysulfides[25]. Functional groups containing nitrogen and/or sulfur also show strong binding and are capable of anchoring polysulfides[26,27]. However, these functional groups are often unstable and it is difficult to control their contents[28]. Because of this, many groups have used polar oxides for chemically adsorbing polysulfides. For instance, MnO$_2$ nanosheets were used to spatially locate and control the deposition of both Li$_2$S/Li$_2$S$_2$ and sulfur by offering an active interface via the thiosulfate intermediate[29]. Silica has also been used as a polysulfide adsorbent, because of its excellent stability and high specific surface area. In conjunction with a polyethylene oxide coating on a separator, self-discharge was increased due to

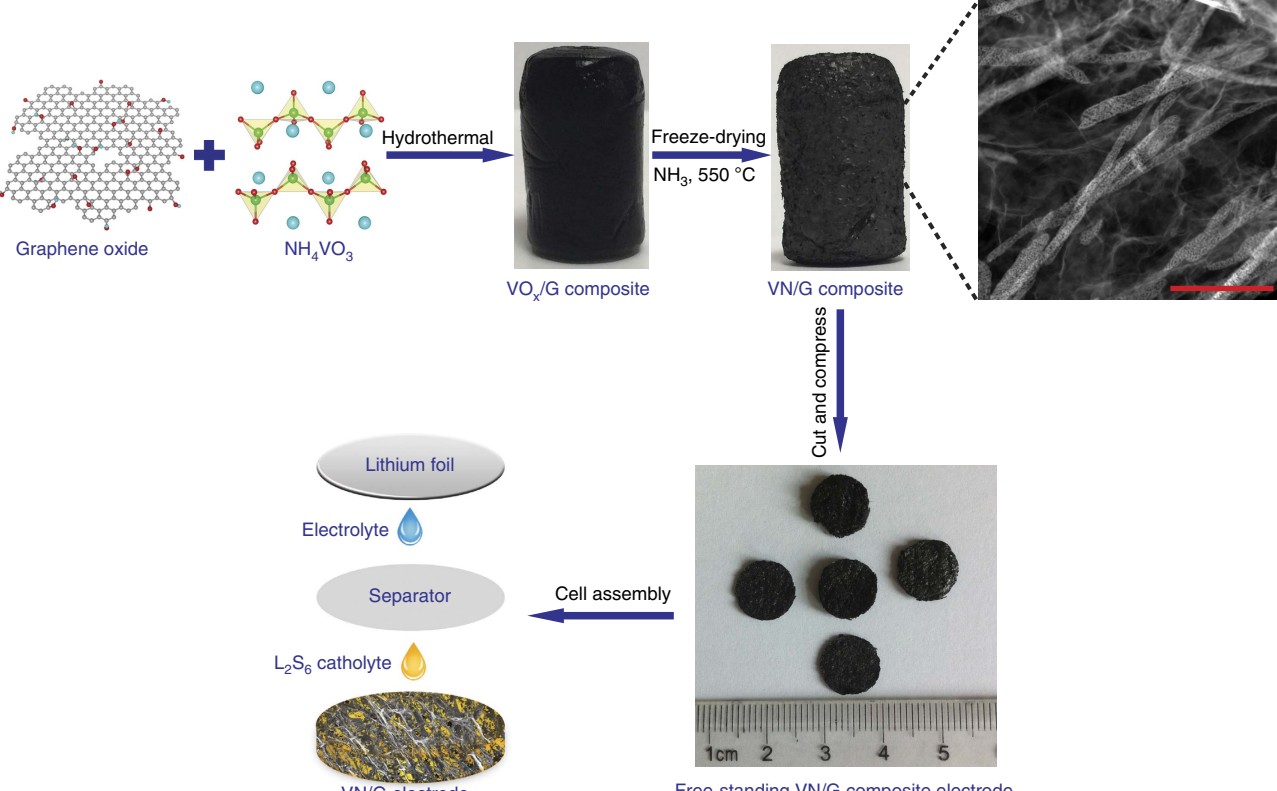

**Figure 1 | Schematic of fabrication process of VN/G composite and cell assembly.** Schematic of the fabrication of a porous VN/G composite and the cell assembly with corresponding optical images of the material obtained. Scale bar, 500 nm.

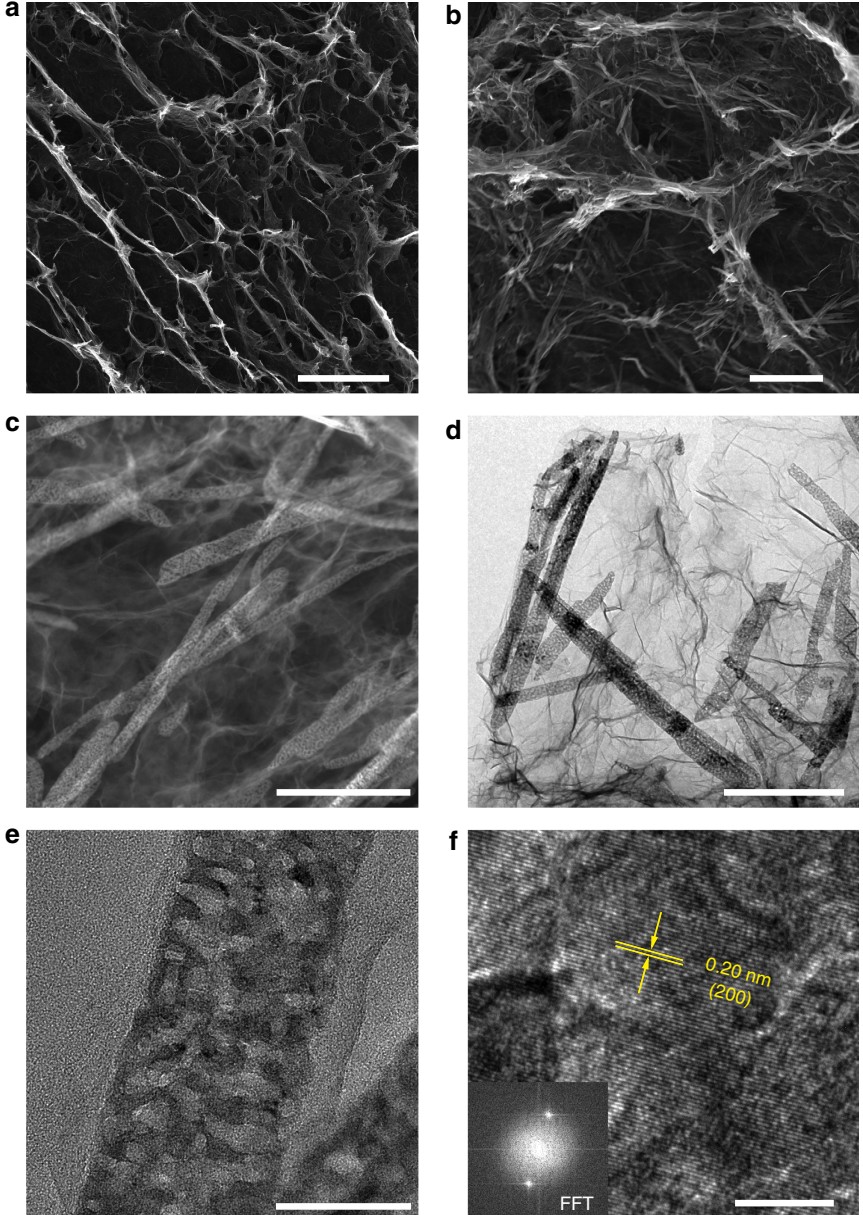

**Figure 2 | Morphology and structural characterization of the VN/G composite.** (**a**) Low-magnification SEM image, (**b**) high-magnification SEM image, (**c**) high-angle annular dark-field (HAADF) STEM image and (**d**,**e**) TEM images of the as-prepared porous VN/G composite. (**f**) High-resolution TEM (HRTEM) image, with inset showing the fast Fourier transform (FFT) pattern. Scale bars, (**a**) 100 µm; (**b**) 2 µm; (**c**) 500 nm; (**d**) 500 nm; (**e**) 50 nm; (**f**) 5 nm.

the strong polysulfide-silica interactions causing polysulfide diffusion from the cathode[30]. Nonetheless, insulating oxides ultimately impede electron transport and interrupt paths for Li ion movement, thus leading to low sulfur utilization and rate capability. It is worth noting that introducing highly conductive polar materials into the sulfur electrode is an effective means of alleviating the above issues. For example, the surface of added metallic $Ti_4O_7$ triggers the reduction of sulfur and oxidation of $Li_2S$ by forming an excellent interface with polysulfides[31]. Similarly, the addition of MXene phase $Ti_2C$ introduces exposed terminal metal sites that bond with sulfur as a result of an interface-mediated reduction[32]. Metal nitrides with a high electrical conductivity can be an ideal anchoring material. A generalized gradient approximation and local density approximation analysis of a series of transition metal

nitrides (TiN, VN, CrN, ZrN and NbN) indicate the metallic behaviour of these materials with no resolved band gap[33]. Among metal nitrides, vanadium nitride (VN) has a number of desirable properties for a potential host materials for sulfur including the following: (1) a strong chemical adsorption for polysulfides that can effectively inhibit the shuttle effect, (2) a high electrical conductivity ($1.17 \times 10^6\,S\,m^{-1}$ at room temperature) (Supplementary Table 1) that is conducive to the electrochemical conversion of adsorbed sulfur species on the surface and (3) catalytic properties similar to the precious metals that may facilitate redox reaction kinetics.

Here we report a highly conductive porous VN nanoribbon/ graphene (VN/G) composite accommodating a suitable amount of $Li_2S_6$ catholyte as the cathode of Li–S batteries without using carbon black and binder. The free-standing three-dimensional

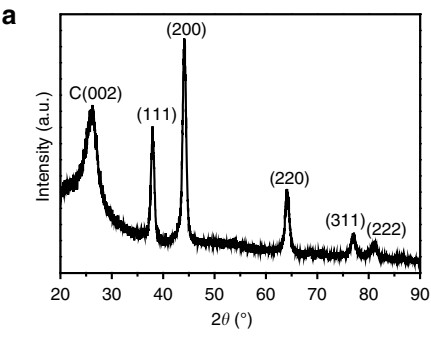

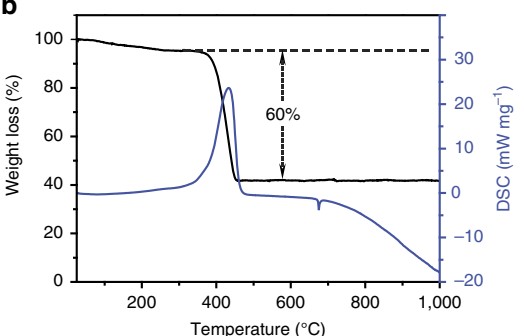

**Figure 3 | Compositional information of the VN/G composite.** (**a**) X-ray diffraction (XRD) pattern and (**b**) thermogravimetric-differential scanning calorimetry (TG-DSC) curve of the VN/G composite.

(3D) interconnected network of the graphene facilitates the transportation of electrons and lithium ions, and the VN not only shows strong chemical anchoring of the polysulfides, but also accelerates the redox reaction kinetics. The anchoring of polysulfides by VN is investigated in a dissolved polysulfide system and further verified by theoretical calculations. The VN/G cathode delivers a high specific capacity of 1,461 mAh g$^{-1}$ at 0.2 C, a Coulombic efficiency approaching 100%, and a high-rate performance of 956 mAh g$^{-1}$ at 2 C.

## Results

**Synthesis and characterization of VN/G composite.** As illustrated in Fig. 1, the synthesis of a porous VN/G composite involves two steps. We first obtained a vanadium oxide/graphene (VO$_x$/G) hydrogel by a hydrothermal method using graphene oxide and NH$_4$VO$_3$ as precursors. VO$_x$ was grown *in situ* on the surface of the graphene oxide and simultaneously assembled into a 3D foam. After immersion in deionized water, the product was subjected to freeze-drying and a VO$_x$/G macrostructure was formed. After annealing in a NH$_3$ atmosphere, the free-standing VN/G composite was obtained. The final product can be cut and pressed into plates for direct use as Li–S battery electrodes without a metal current collector, binder and conductive additive.

The morphology and microstructure of the VN/G composite were characterized by scanning electron microscopy (SEM) and transmission electron microscopy (TEM) as shown in Fig. 2. SEM images reveal that 3D interconnected network of VN nanoribbons and reduced graphene oxide (RGO) sheets. Numerous voids, several micrometres in size, are able to hold a large amount of sulfur and provide good penetration of electrolyte (Fig. 2a,b). This skeleton structure not only enhances the electron and lithium ion transportation but also accommodates the volume expansion of sulfur. The elemental mappings of vanadium, nitrogen, carbon and oxygen further reveal the hybrid

structure of the VN/G composite (Supplementary Fig. 1). To see this more clearly, we then characterized the structure using a high-angle annular dark-field scanning TEM (STEM) and TEM in Fig. 2c–e. The VN nanoribbons are typically 50–100 nm wide and 1–2 μm long. Compared with the product before annealing in NH$_3$ (Supplementary Fig. 2), VN nanoribbons contains a large number of mesopores ranging from 10 to 30 nm in diameter, which are beneficial for both the ion transportation and the adsorption of polysulfides in the electrochemical process. A representative high-resolution TEM image and the fast Fourier transform pattern are also shown in Fig. 2f, revealing lattice fringes with a spacing of 0.20 nm, which is in agreement with spacing of the (200) plane of VN. The graphene in the VN/G composite provides a supporting framework to prevent the aggregation of the VN nanoribbons.

The crystal structure of the 3D VN/G composite was further examined by X-ray diffraction (Fig. 3a). The major peaks are assigned to cubic VN (JCPDS card number 73-0528) with a wide peak around 26° corresponding to graphene stacking. Thermogravimetric-differential scanning calorimetry analysis suggested that the VN content was 30% (Fig. 3b). The specific surface area of the VN/G was 37 m$^2$ g$^{-1}$ with mesopores 18 nm in diameter (Supplementary Fig. 3), which is consistent with the TEM observation. In contrast, the specific surface area of the RGO was as high as 296 m$^2$ g$^{-1}$ (Supplementary Fig. 4).

**The electrochemical performance of VN/G cathodes.** A series of electrochemical measurements were carried out to evaluate the performance of the VN/G cathode. In the cell assembly process, Li$_2$S$_6$ catholyte was directly added to VN/G (Fig. 1). The final areal sulfur loading of the electrode was 3 mg cm$^{-2}$. Typical cyclic voltammetry (CV) profiles for the RGO and VN/G electrodes were obtained within a potential window of 1.7–2.8 V at a scan rate of 0.1 mV s$^{-1}$ (Fig. 4a), both showing two cathodic peaks and two anodic peaks. The two representative cathodic peaks can be attributed to the reduction of sulfur to long-chain lithium polysulfides (Li$_2$S$_x$, $3 \leq x \leq 8$) at the higher potential and the formation of insoluble short-chain Li$_2$S$_2$/Li$_2$S at the lower potential. When scanning back, the anodic peaks corresponded to the oxidation of Li$_2$S/Li$_2$S$_2$ to polysulfides and then to sulfur. It is interesting to note that the reduction peaks with the VN/G cathode (2.0 and 2.35 V) appeared at higher potentials than those with the RGO cathode (1.88 and 2.24 V). The distinguishable positive shift in the reduction peaks and negative shift in the oxidation peaks of the VN/G cathode indicate the improved polysulfide redox kinetics by VN. According to recent reports, Pt as an electrocatalyst can help to convert polysulfide deposits back to soluble long-chain polysulfide and hence enhance reaction kinetics and retain high Coulombic efficiency, and the catalytic activities of VN resemble those of noble metal Pt[34,35]. These results suggest that VN has similar catalytic activity to that of precious metals, which can improve the redox reaction kinetics. Galvanostatic charge/discharge tests (Fig. 4b) were further performed at a constant current rate of 0.2 C (based on the mass of sulfur in the cell, 1 C = 1,675 mA g$^{-1}$). The charge–discharge profiles of VN/G consist of two discharge plateaus at 2.35 and 2.05 V, and two charge plateaus between 2.2 and 2.45 V, respectively, which are in agreement with the CV curves. The plateaus were longer and flatter with a higher capacity and a lower polarization than those using the RGO electrode, suggesting a kinetically efficient reaction process. Figure 4c shows the cycling performance of the VN/G and RGO cathodes. The VN/G cathode delivered an excellent initial discharge capacity of 1,471 mAh g$^{-1}$ and, more importantly, it was able to maintain a stable cycling performance with a Coulombic

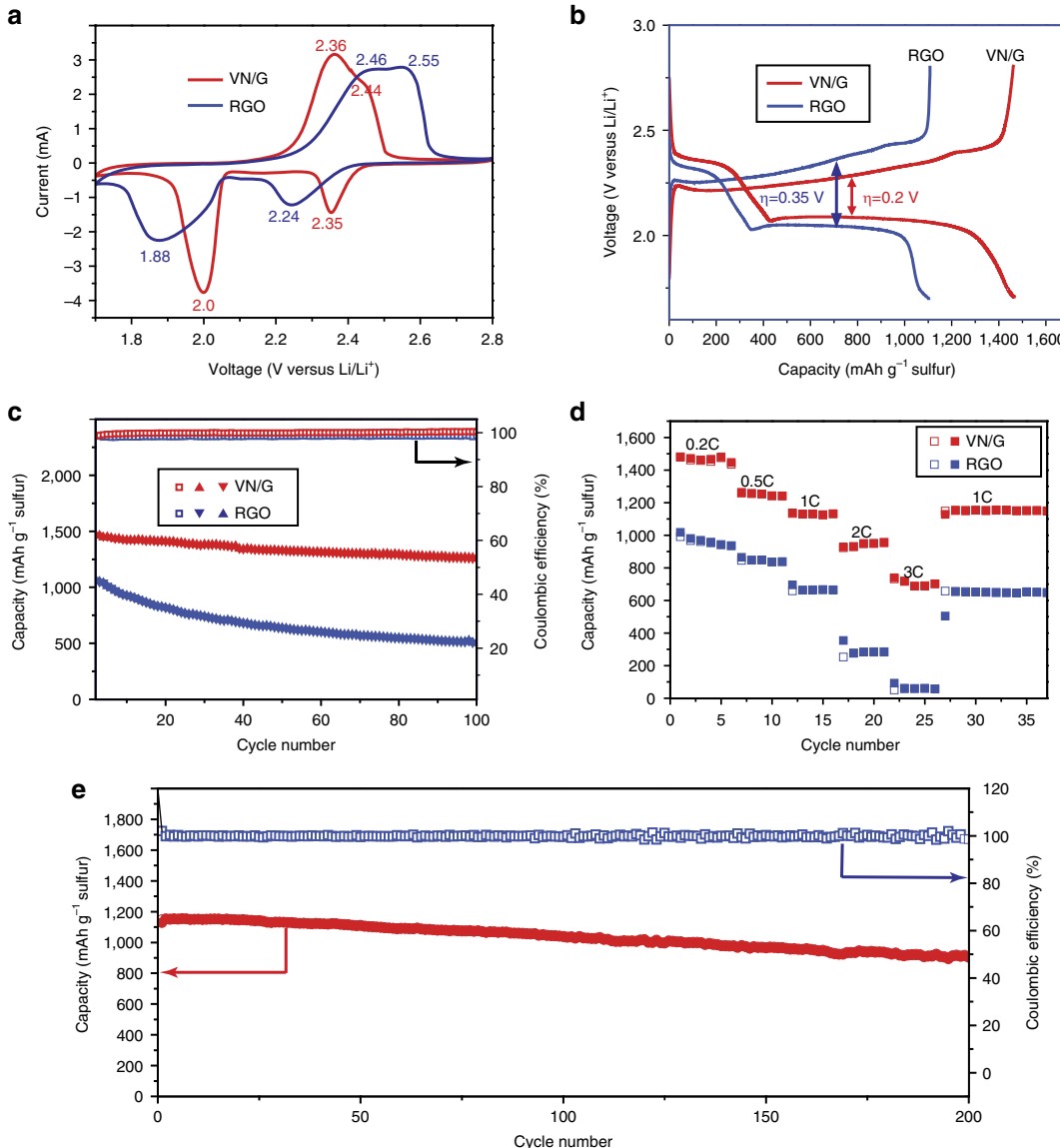

**Figure 4 | Electrochemical performances of VN/G and RGO cathodes.** (**a**) CV profiles of the VN/G and RGO cathodes at a scan rate of $0.1\,mV\,s^{-1}$ in a potential window from 1.7 to 2.8 V. (**b**) Galvanostatic charge–discharge profiles of the VN/G and RGO cathodes at 0.2 C. (**c**) Cycling performance and Coulombic efficiency of the VN/G and RGO cathodes at 0.2 C for 100 cycles. (**d**) Rate performance of the VN/G and RGO cathodes at different current densities. (**e**) Cycling stability of the VN/G cathode at 1 C for 200 cycles.

efficiency above 99.5% for 100 charge–discharge cycles at 0.2 C, indicating that dissolution of polysulfides into the organic electrolyte was effectively mitigated in the VN/G electrode. The LiNO$_3$ additive in the electrolyte also has a positive effect on the Coulomb efficiency and cyclic performance of Li–S batteries[36]. It was also confirmed that the VN/G host contributed almost nothing to the measured capacity (Supplementary Fig. 5). In contrast, the RGO cathode showed a lower discharge capacity of $1,070\,mAh\,g^{-1}$ in the initial cycle and rapid capacity decay with a capacity retention of 47% after 100 cycles, implying low sulfur utilization with severe polysulfide dissolution into the electrolyte. In the electrochemical impedance spectroscopy measurements (Supplementary Fig. 6), the Nyquist plots obtained consist of two parts, a semicircle in the high-frequency region representing the charge transfer resistance and a straight line in the low-frequency region associated with the mass transfer process. The VN/G cathode has a smaller resistance ($28\,\Omega$) than that of the RGO cathode ($95\,\Omega$), which can be explained by enhanced interfacial

affinity between VN and polysulfides, and the high electrical conductivity of metal nitrides comparable to their metal counterparts, as shown in Supplementary Table 1. In addition, the VN/G composite also exhibits an electrical conductivity of $\approx 1,150\,S\,m^{-1}$ measured by the four-point probe method, which is over four times larger than that of RGO (about $240\,S\,m^{-1}$), even though RGO contains doping nitrogen (about 4.6%) after NH$_3$ annealing (Supplementary Fig. 7). Although N-doped graphene can improve the performance of Li–S batteries, but the electrochemical performance of VN/G composite electrode was much better than that of RGO electrode in the same condition. As shown in Fig. 4d, when the electrode was cycled at different rates of 0.2 C, 0.5 C, 1 C, 2 C and 3 C, the cell was able to deliver discharge capacities of 1,447, 1,241, 1,131, 953 and $701\,mAh\,g^{-1}$, respectively. In contrast, the RGO electrode exhibited lower discharge capacity and poorer stability under the same conditions. Moreover, a stable discharge capacity of $1,148\,mAh\,g^{-1}$ was recovered as soon as the current density was

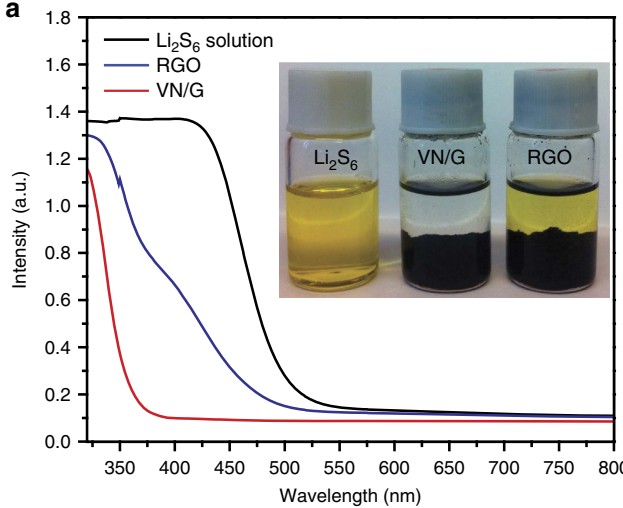

**a**

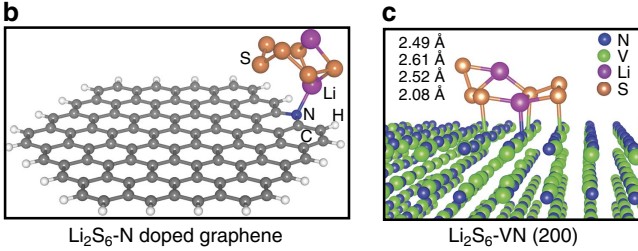

**b**

Li$_2$S$_6$-N doped graphene

**c**

2.49 Å
2.61 Å
2.52 Å
2.08 Å

Li$_2$S$_6$-VN (200)

**Figure 5 | Demonstration of the strong interaction of VN/G composite with polysulfides.** (**a**) Ultraviolet/visible absorption spectra of a Li$_2$S$_6$ solution before and after the addition of RGO and VN/G. Inset image shows a photograph of a Li$_2$S$_6$ solution before and an 2 h after the addition of graphene and VN/G. (**b**) Side view of a Li$_2$S$_6$ molecule on a nitrogen-doped graphene surface, the binding energy between Li$_2$S$_6$ and pyridinic N-doped graphene is calculated to be 1.07 eV. (**c**) Side view of a Li$_2$S$_6$ molecule on VN (200) surface, the binding energy between Li$_2$S$_6$ and VN is calculated to be 3.75 eV.

restored to 1 C. Figure 4e shows the long-term cyclability of VN/ G electrode at 1 C, indicating an excellent cycling stability. The initial capacity was as high as 1,128 mAh g$^{-1}$ and retained 81% of the initial capacity (917 mAh g$^{-1}$) after 200 cycles. Although higher polarization occurred in the electrodes at higher rates due to slower dynamics of sulfur, the charge–discharge profiles still consist of two plateaus even at a very high current density (Supplementary Fig. 8). In contrast, the VO$_x$/G electrode displayed rapid capacity decay and low Coulombic efficiency (about 93% after 100 cycles), which probably resulted from the low conversion efficiency of polysulfides adsorbed on non-conductive VO$_x$ surfaces (Supplementary Fig. 9). The excellent electrochemical performance of the VN/G cathode can be attributed to the following factors. First, the porous VN host provides a polar surface and a strong chemical interaction with polysulfides, effectively inhibiting the shuttle effect. Second, the high electrical conductivity of VN enhances redox electron transfer and reduces interfacial impedance, and accelerates the polysulfide conversion. Third, VN has similar catalytic activity to that of the precious metals, which improves the redox reaction kinetics.

## Discussion

To verify the strong anchoring of VN for polysulfides, as shown in Fig. 5a, we compared the polysulfide adsorption ability of

RGO and the VN/G composite, after adding 20 mg of their powders to Li$_2$S$_6$ solution for 2 h. The VN/G completely decoloured the polysulfide solution, whereas the solution containing RGO remained the same bright yellow colour. Ultraviolet/visible absorption measurements were also made to investigate the concentration changes of Li$_2$S$_6$ solutions after adding RGO or VN/G. It can be clearly seen that the absorption peak of Li$_2$S$_6$ in the visible light range apparently disappeared after adding VN/G, but remained after adding RGO (Fig. 5a). This difference suggests strong adsorption of Li$_2$S$_6$ molecules to polar VN, owing to ionic bonding of V–S. The surface compositions of VN/G composite were measured by X-ray photoelectron spectroscopy survey spectra indicates that the surface of the VN also contains small amounts of V–N–O and V–O bonds, which have a high affinity for polysulfides (Supplementary Fig. 10)[37]. The strong interaction between VN and lithium polysulfides was further verified by an evaluation of the binding energies between Li$_2$S$_6$ and VN based on density functional theory calculations (Supplementary Note 1). As shown in the Supplementary Fig. 7, the pyridinic-N is the dominant dopant in N-doped graphene synthesized in this work. For comparison, the binding energy between Li$_2$S$_6$ and pyridinic N-doped graphene was considered, and it has been reported to be 1.07 eV[38]. In contrast, the binding energy between Li$_2$S$_6$ and VN was calculated to be much larger (3.75 eV). This is mainly due to the much stronger polar–polar interactions between Li$_2$S$_6$ and VN than those between Li$_2$S$_6$ and pyridinic N-doped graphene. In comparison with the case of Li$_2$S$_6$ on pyridinic N-doped graphene (Fig. 5b), the strong polar–polar interaction between Li$_2$S$_6$ and VN results in an obvious deformation of the Li$_2$S$_6$ molecule (Fig. 5c), forming three S–V and one Li–N bonds. The bond lengths of these S–V (2.49–2.61 Å) and Li–N (2.08 Å) bonds are very close to the corresponding bond lengths in bulk VS (2.42 Å) and LiNH$_2$ (2.06 Å), respectively[39,40]. These results clearly show the good affinity and strong chemical anchoring of polar VN for polysulfides. In addition, the non-polarity of graphene in the VN/G composite can also be beneficial for the redeposition of the charging product sulfur. The hetero-polar VN/G electrodes provide both polar (VN) and non-polar (graphene) platforms to facilitate the binding of solid Li$_x$S and sulfur species to the electrodes. STEM elemental mapping was performed to track the sulfur distribution in the VN nanoribbons after cycling. The high-angle annular dark-field STEM image and corresponding elemental maps of vanadium, nitrogen and sulfur show that the sulfur species were uniformly distributed and strongly adsorbed on the surface of the VN nanoribbons (Fig. 6). This result verifies the experimental observations and corresponding theoretical calculations.

In summary, we have used a 3D highly conductive porous VN/G composite to solve the shuttle effect in Li–S batteries. This composite combines the advantages of both graphene and VN. The 3D free-standing structure composed of a graphene network facilitates electron and ion transportation, but is also beneficial to electrolyte absorption. In addition, VN showed a strong anchoring effect for polysulfides and its high conductivity also accelerated the polysulfide conversion. The VN/G electrode exhibited excellent specific capacity with a Coulombic efficiency reaching >99% compared with the RGO electrodes. We believe that other highly conductive metal nitrides can also be used for high-energy Li–S batteries and our design opens a new direction of the electrochemical use of transition metal nitrides for energy storage.

## Methods

**Preparation of a 3D porous VN/G composite.** The VN/G composites were prepared using hydrothermal method, according to the previously reported

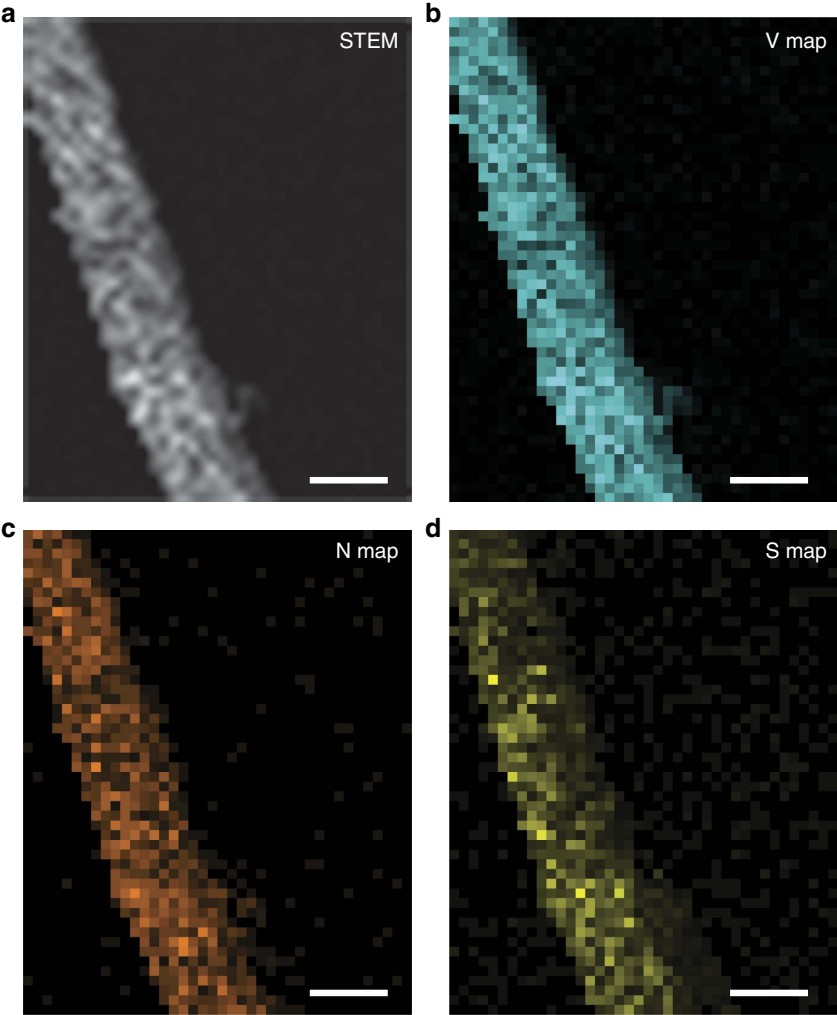

**Figure 6 | Sulfur distribution in the VN nanoribbons after cycling.** (**a**) STEM image of a VN nanoribbon after cycling with the corresponding elemental maps of (**b**) vanadium, (**c**) nitrogen and (**d**) sulfur. Scale bars, 100 nm.

procedure[41]. Specifically, 0.05 g $NH_4VO_3$ was dissolved in a mixture of 45 ml water and 5 ml ethanol, followed by slowly adding drops of HCl (2 M) to adjust the pH of the solution to 2–3. Next, 30 ml of a graphene oxide suspension (5 mg ml$^{-1}$) was added to the solution under continuous stirring. The mixture was then transferred to a 100 ml Teflon-lined autoclave, which was heated to 180 °C where it was maintained for 24 h. The as-prepared sample was rinsed with deionized water several times followed by freeze-drying for 2 days. Finally, the obtained product was heated at 550 °C for 3 h in an $NH_3$ (30 s.c.c.m.) atmosphere. For comparison, a 3D RGO structure was prepared following the same procedure. The 3D porous $VO_x$/G composite was also synthesized using a process similar to that for the synthesis of VN/G composite, except that the atmosphere of the heat treatment was changed from ammonia to argon.

**Preparation of the $Li_2S_6$ solution.** Sulfur and $Li_2S$ at a molar ratio of 5:1 were added to an appropriate amount of 1,2-dimethoxyethane and 1,3-dioxolane by vigorous magnetic stirring at 50 °C until the sulfur was fully dissolved.

**Polysulfide adsorption test.** A solution with a $Li_2S_6$ concentration of 50 mmol l$^{-1}$ (calculated based on sulfur content) was used. Twenty milligrams of VN/G and RGO powder were separately added to 2.0 ml of $Li_2S_6$ solution and the mixtures were stirred to obtain thorough adsorption. A blank glass vial was also filled with the same $Li_2S_6$ solution as a comparison.

**Preparation of sulfur electrodes.** A VN/G composite was cut and compressed into 1.5 mg VN/G electrode. Next, inside an Argon-filled glovebox, 30 μl $Li_2S_6$ catholyte equal to 1.92 mg of sulfur and 60 μl of electrolyte was used to form the sulfur electrode. The final areal sulfur loading of the electrode was determined about 3 mg cm$^{-2}$.

**Materials characterization.** The morphology and structure of the materials were characterized using a SEM (FEI Nova NanoSEM 450, 15 kV). TEM imaging was performed on a FEI CM120 microscope. High-resolution TEM images, STEM images and energy dispersive X-ray spectroscopy (EDX) elemental maps were obtained on a FEI Tecnai F20 microscope equipped with an Oxford EDX analysis system with an acceleration voltage of 200 kV. X-ray diffraction patterns were obtained on a Rigaku diffractometer (Cu $K_\alpha$, $\lambda = 0.154056$ nm). Thermogravi-metric-differential scanning calorimetry analysis (TGA) was performed with a NETZSCH STA 449 C thermo balance in air with a heating rate of 10 °C min$^{-1}$ from room temperature to 1,000 °C. The X-ray photoelectron spectroscopy measurements were carried out in an ultra-high vacuum ESCALAB 250 set-up equipped with a monochromatic Al $K\alpha$ X-ray source (1486.6 eV; anode operating at 15 kV and 20 mA). Ultraviolet/visible absorption spectroscopy analysis (Cary 5000) was performed to evaluate the polysulfide adsorption capability of RGO and VN/G. The electrical conductivities were measured by a standard four-point-probe resistivity measurement system (RTS-9, Guangzhou, China). $N_2$ adsorption/desorption isotherms were determined using a Micromeritics ASAP2020M instrument. Before the measurements, the samples were degassed at 200 °C until a manifold pressure of 2 mm Hg was reached. The surface area and pore size distribution were determined based on the Barrett–Joyner–Halenda method.

**Electrochemical measurements.** Stainless steel coin cells (2,032-type) were assembled inside an Ar-filled glovebox. The electrolyte was lithium bis-trifluoromethaesulphonylimide (99%, Acros Organics, 1 M) dissolved in 1,3-dioxolane (99.5%, Alfa Asea) and 1,2-dimethoxyethane (99.5%, Alfa Aesar) (1:1 ratio by volume) with 0.2 M lithium nitrate ($LiNO_3$, 99.9%, Alfa Aesar) as the additive. Lithium metal foil was used as the anode and Celgard 2400 as the separator. A Landian multichannel battery tester was used to perform electro-chemical measurements. The charge-discharge voltage range was 1.7–2.8 V. The CV and the electrochemical impedance spectroscopy measurements were performed on a VSP-300 multichannel workstation.

**Data availability.** The authors declare that the data supporting the findings of this study are available within the article and its Supplementary Information files. All other relevant data supporting the findings of this study are available from the corresponding author on request.

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

## Acknowledgements

We acknowledge financial support from MOST (2016YFA0200100, 2014CB932402 and 2016YFB0100100) and the National Science Foundation of China (numbers 51525206, 51521091, 51472249, 51372253, 51272051 and U1401243), Youth Innovation Promotion Association of the Chinese Academy of Sciences (number 2015150), the Natural Science Foundation of Liaoning province (number 2015021012), the Institute of Metal Research (number 2015-PY03) and 'Strategic Priority Research Program' of the Chinese Academy of Sciences (XDA09010104), the Key Research Program of the Chinese Academy of Sciences (grant number KGZD-EW-T06) and the CAS/SAFEA International Partnership Program for Creative Research Teams. The theoretical calculations were performed on TianHe-1(A) of National Suercomputer Center in Tianjin. We thank Dr Wei Lv and Professor Quanhong Yang for helping in experiments.

## Author contributions

Z.S. and F.L. designed the research. Z.S. conducted the electrochemical experiments and characterization of materials, and J.Z. prepared the materials. L.Y. performed density functional theory calculations. G.H. and R.F. contributed to the discussion of the results. Z.S., J.Z., H.-M.C. and F.L. wrote the paper. All the authors commented on and revised the manuscript.
