## [Peer Review File · Nature Communications]

Reviewers' comments:

Reviewer #1 (Remarks to the Author):

This paper reports porous VN/graphene composite as chemical anchor of polysulfides. The authors report that VN effectively acts as a chemical anchor for improving cell performance of Li-S batteries using polysulfide chatholyte.

This would be the first report that using VN as a chemical anchor of polysulfides. The VN/G composite shows long cycling performance and relatively high rate performance. The result shown in this paper will be useful for the researchers of Li-S batteries.

On the other hand, this paper includes many unsupported discussions. To be accepted in Nat. Commun., the authors should revise the manuscript based on following comments.

1. The material characterization seems to be not enough.
2. The density of electrode (VN/G composite) seems to be very low. Thus, the cell needs a large volume ratio of electrolyte. Is the porous structure shown in this paper effective form?
3. The scientific novelty in this paper should be shown more clearly. e.g. The scientific discussion about catalytic properties is not enough.

Reviewer #2 (Remarks to the Author):

It is quite impressive that the monolith of graphene/VN exhibits very strong adsorption/absorption capability towards polysulfide species, considering its relatively low specific surface area. The loading of the active mass in the cells is good, and the capacity, cycling performance as well as coulombic efficiency support the argument that VN is a strong anchoring composition in the cathode toward polysulfide. The preparation of a monolith by a hydrothermal reaction is innovative, followed by a nitrodition reaction under NH₃. The article is well written, which can be accepted after a minor revision.

Annealing under NH₃ would turn GO into N-doped RGO. Are there data of XPS to reveal the composition of this possibility? Did the authors observe mass loss of GO during NH₃ annealing?

As LiNO₃ is employed in the electrolyte, the high coulombic efficiency may not be directly attributed to the anchoring effect. Have you tried the case without LiNO₃?

Please provide the theoretical conductivity information of VN in the revised text. What is the conductivity of the Graphene/VN composite?

Figure 1a, the schematic for the GO should be revised to reflect the existence of oxygen containing groups.

Reviewers' comments:

Reviewer #1 (Remarks to the Author):

The manuscript has been well revised.

Reviewer #2 (Remarks to the Author):

The authors have addressed the comments very well. It can be accepted.

Reviewer #3 (Remarks to the Author):

Sun et al. report the application of a VN/graphene composite as a host material for a catholyte Li-S cell. The highlighted scientific advance was the use of a conductive metal nitride material and its catalytic properties towards polysulfide redox. However, there are some concerns that prevent recommendation for publishing in Nature Commun.

1) There have been previous reports on using conductive metal nitrides, TiN (Adv. Mater. 2016, DOI: 10.1002/adma.201601382, J. Mater. Chem. A, DOI: 10.1039/C6TA07411A), which shows the benefits of adding or using nitrides in sulfur host materials. This may compromise the significance of this report

2) There have been quite a few studies on using oxides. The authors have to demonstrate the true advances of using nitrides over oxides, or at least compare the performances with the oxides-based hosts, e.g VOx, in order to highlight the significance of using VN.

3) The use of 70 wt% of porous graphene sponge requires a large amount of electrolyte, >30 ul/mg sulfur as the authors report. This lowers the volumetric energy density. Even still, there have been reports of using graphene based catholyte cells that apply 8.5 mg cm⁻² of sulfur loading or more (Nature Commun. 2015, 6, 7760). Good performance at higher loading have to be demonstrated in this manuscript.

4) Some other details that can help improve the manuscript.

1. The surface area of rGO needs to be reported and compared with VN-rGO

2. In Figure S5, what is the condition of the cell for EIS? Pristine or in discharged/ charged states? This is important for claiming the affinity between VN and polysulfides.

3. In Figure 4d, the rate capability of rGO cell has to be compared directly.

4. In Figure 5, the binding energy of VN with Li₂S₆ has to be compared with N-doped graphene, instead of plain graphene, in order to correlate with the experiments.

5. On line 251, page 13, the bonding length in original literature needs to be provided for comparison.

REVIEWERS' COMMENTS:

Reviewer #3 (Remarks to the Author):

The authors have now properly addressed the questions and highlighted the major contribution of this paper to the Li-S community. It can be accepted at this stage.

Response to Reviewer 1

Overall comments: This paper reports porous VN/graphene composite as chemical anchor of polysulfides. The authors report that VN effectively acts as a chemical anchor for improving cell performance of Li-S batteries using polysulfide catholyte. This would be the first report that using VN as a chemical anchor of polysulfides. The VN/G composite shows long cycling performance and relatively high rate performance. The result shown in this paper will be useful for the researchers of Li-S batteries. On the other hand, this paper includes many unsupported discussions. To be accepted in Nat. Commun., the authors should revise the manuscript based on following comments.

We thank this referee for the very positive comments and valuable suggestions.

Question 1: The material characterization seems to be not enough.

Response 1: Thanks for the referee's suggestion. In the original manuscript, the VN/G composite and RGO were characterized by the common characterization methods such as SEM, TEM, STEM, XRD, TG-DSC, nitrogen adsorption-desorption and so on. To better understand the structure and properties of this material, we have added SEM-EDX mappings (as Supplementary Figure 1), XPS spectra (as Supplementary Figure 4 and Figure7) and electrical conductivity data (as Supplementary Table 1), and also added corresponding discussion in the revised manuscript.

Question 2: The density of electrode (VN/G composite) seems to be very low. Thus, the cell needs a large volume ratio of electrolyte. Is the porous structure shown in this paper effective form?

Response 2: The density of the VN/G composite macroform is 276 mg cm^{-3} measured by a balance (METTLER TOLEDO XS205) equipped with accessories for the density determination by the Archimedes principle, which is more than two times higher than that of graphene macrostructure prepared by the same freeze-drying method ($<100 \text{ mg cm}^{-3}$ in most cases, Energy Environ. Sci., 2015, 8, 1390 and Sci. Rep. 2013, 3, 2975). Controlling the drying process is a simple but effective way to simultaneously tailor

the structures and properties of graphene-based macrostructure, such as pore structure, density, mechanical strength, conductivity etc. According to the literature (Sci. Rep. 2013, 3, 2975 and J. Phys. Chem. Lett. 2015, 6, 658-668.), the density and porosity of graphene-based macrostructure materials can be tuned (the density as high as 1.58 g cm⁻³) by an evaporation-induced drying method of a graphene hydrogel. Therefore, the density of our VN / G composites can be adjusted according to different requirements. In addition, a large packing pressure (about 7MPa) during the assembly of a coin cell can compact the VN/G electrode, further increasing the density of the electrodes and reducing the amount of electrolyte used. Many studies¹⁻⁵ also show that graphene-based 3D macroscopic electrode exhibit a highly porous network structure and abundant electrically conducting pathways, which can be cut and pressed into pellets to be directly used as electrode without using a metal current collector, binder, and conductive additive.

- [1] L. W. Ji, P. Meduri, W. Agubra, X. C. Xiao, M. Alcoutlabi, Graphene-based nanocomposites for energy storage, *Adv. Energy Mater.* **2016**, DOI:10.1002/aenm.201502159.
- [2] Q. Shi, Y. Cha, Y. Song, J. Lee, C. Zhu, X. Li, M. K. Song, D. Du, Y. H. Lin, 3D graphene-based hybrid materials: synthesis and applications in energy storage and conversion, *Nanoscale* **2016**,8, 15414.
- [3] Z. B. Lei, J. T. Zhang, L. L. Zhang, N. A. Kumar, X. S. Zhao, Functionalization of chemically derived graphene for improving its electrocapacitive energy storage properties, *Energy Environ. Sci.* **2016**, 9, 1891.
- [4] Y. Shao, M. F. El-Kady, L. J. Wang, Q. Zhang, Y. Li, H. Z. Wang, M. F. Mousavi, R. B. Kaner, Graphene-based materials for flexible supercapacitors, *Chem. Soc. Rev.* **2015**, 44, 3639.
- [5] S. Han, D. Wu, S. Li, F. Zhang, X. L. Feng, Porous graphene materials for advanced electrochemical energy storage and conversion devices, *Adv. Mater.* **2014**, 26, 849.

Question 3: *The scientific novelty in this paper should be shown more clearly. e.g. The scientific discussion about catalytic properties is not enough.*

Response 3: Thank the referee for constructive suggestion. In order to highlight the scientific novelty of this manuscript, we have added the discussion about catalytic

properties of VN as follows:

According to recent reports, the presence of electrocatalyst (Pt or Ni) helps to convert the polysulfide deposits back to soluble long-chain polysulfide and hence enhances reaction kinetics and retains high Coulombic efficiency (*J. Am. Chem. Soc.* **137**, 11542-11545 (2015)). The deposition of insulating polysulfide on electrode can impede the electron transfer at the electrode/electrolyte interface and results in an increase of internal resistance. It is known that Pt is promising but expensive as an electrocatalyst to convert short-chain to long-chain lithium polysulfides efficiently in a kinetically facile manner during charging. The catalytic properties of metal nitrides have been the subject of many experimental and theoretical investigations. In many instances the catalytic activities of VN resemble those of noble metals like Pt. Recent research shows that VN has an electrocatalytic activity similar to Pt (*Sci. Rep.* **5**, 11351 (2015)). In our experiments, we found that the reduction peaks with the VN/G cathode (2.0 and 2.35V) appeared at higher potentials than those with the reduced graphene oxide cathode (1.88 and 2.24V) in the cyclic voltammetry profiles. We therefore conclude that the improved polysulfide redox kinetics may be derived from the high electrical conductivity and catalytic activity of VN.

We have added the sentence “According to recent reports, Pt as an electrocatalyst can help to convert polysulfide deposits back to soluble long-chain polysulfide and hence enhance reaction kinetics and retain high Coulombic efficiency, and the catalytic activities of VN resemble those of noble metal Pt. These results suggest that VN has similar catalytic activity to that of precious metals, which can improve the redox reaction kinetics.” after the sentence “The distinguishable positive shift in the reduction peaks and negative shift in the oxidation peaks of the VN/G cathode indicate the improved polysulfide redox kinetics by VN.” and labeled in the manuscript. We have also added the literature (*J. Am. Chem. Soc.* **137**, 11542-11545 (2015) and *Sci. Rep.* **5**, 11351 (2015)) in revised manuscript as reference 34 and 35, respectively.

Response to Reviewer 2

Overall comments: It is quite impressive that the monolith of graphene/VN exhibits very strong adsorption/absorption capability towards polysulfide species, considering its relatively low specific surface area. The loading of the active mass in the cells is good, and the capacity, cycling performance as well as coulombic efficiency support the argument that VN is a strong anchoring composition in the cathode toward polysulfide. The preparation of a monolith by a hydrothermal reaction is innovative, followed by a nitrodition reaction under NH₃. The article is well written, which can be accepted after a minor revision.

We thank this referee for the very positive comments and valuable suggestions.

Question 1: *Annealing under NH₃ would turn GO into N-doped RGO. Are there data of XPS to reveal the composition of this possibility? Did the authors observe mass loss of GO during NH₃ annealing?*

Response 1: A large number of literature⁶⁻¹² show that the heat treatment of graphene oxide under ammonia gas is an effective method to obtain nitrogen doped reduced graphene oxide, which is consistent with the opinion of the reviewers. During the annealing process, NH₃ reacts with certain oxygen functional groups in the as-prepared graphene oxide to form C–N bonds. Also, atomic N decomposed from NH₃ can combine with defects sites of graphene oxide, contributing to the formation of stable C–N bonding against high temperature. As suggested by the reviewer, we performed XPS measurements of reduced graphene oxide annealing under NH₃. As shown in the figure below (also see Figure S7 in the supporting information), the C1s spectrum consists of peaks at 284.6, 285.2, 285.8, 286.7, and 289.0, attributed to the C=C, C-OH, C-N, C-O-C and C=O groups, respectively. The N 1s spectrum, ranging from 394 to 408 eV, comprises peaks corresponding to pyridine-like and pyrrolic-like nitrogen atoms. The atomic concentration of N of the N-doped reduced graphene oxide is 4.6% determined from the full-range XPS spectrum. The weight of the graphene oxide is also reduced during the ammonia treatment process due to the large loss of the oxygenated groups of the graphene oxide. By measuring the mass change

of the graphene oxide samples before and after NH₃ treatment, the mass loss of graphene oxide during NH₃ annealing was about 16%.

High-resolution C1s and N1s XPS spectra of the reduced graphene oxide

We have added the sentence “In addition, the VN/G composite also exhibits an electrical conductivity of $\approx 1150 \text{ S m}^{-1}$ measured by the four-point probe method, which is over 4 times larger than that of RGO (about 240 S m^{-1}), even though RGO contains doped nitrogen (about 4.6%) after NH₃ annealing (Supplementary Fig.7). Although N doped graphene can improve the performance of Li-S batteries, but the electrochemical performance of VN/G composite electrode was much better than that of RGO electrode in the same condition.” before the sentence “As shown in Figure 5d, when the electrode was cycled at different rates of 0.2 C, 0.5 C, 1 C, 2 C and 3 C” and labeled in the manuscript.

- [6] Z. S. Wu, W. C. Ren, L. Xu, F. Li, H. M. Cheng, Doped graphene sheets as anode materials with superhigh rate and large capacity for lithium ion batteries *ACS Nano* **2011**, 5, 5463.
- [7] S. B. Yang, L. J. Zhi, K. Tang, X. L. Feng, J. Maier, K. Müllen, Efficient synthesis of heteroatom (N or S)-doped graphene based on ultrathin graphene oxide-porous silica sheets for oxygen reduction reactions, *Adv. Funct. Mater.* **2012**, 22, 3634.
- [8] H. F. Huang, G. S. Luo, L. Q. Xu, C. L. Xu, Y. M. Tang, S. L. Tang, Y. W. Du, NH₃ assisted photoreduction and N-doping of graphene oxide for high performance electrode materials in supercapacitors, *Nanoscale*, **2015**, 7, 2060.
- [9] G. V. Bianco, M. Losurdo, M. M. Giangregorio, P. Capezzuto and G. Bruno, Exploring

and rationalising effective n-doping of large area CVD-graphene by NH_3 , *Phys.Chem.Chem.Phys.*, **2014**, *16*, 3632.

- [10] T. F. Yeh, S. J. Chen, H. Teng, Synergistic effect of oxygen and nitrogen functionalities for graphene-based quantum dots used in photocatalytic H_2 production from water decomposition, *Nano Energy* **2015**, *12*, 476.
- [11] N. W. Pu, Y. Y. Peng, P. C. Wang, C. Y. Chen, J. N. Shi, Y. M. Liu, M. D. Ger, C. L. Chang, Application of nitrogen-doped graphene nanosheets in electrically conductive adhesives, *Carbon* **2014**, *67*, 449.
- [12] T. V. Khai, H. G. Na, D. S. Kwak, Y. J. Kwon, H. Ham, K. B. Shim, H. W. Kim, Influence of N-doping on the structural and photoluminescence properties of graphene oxide films, *Carbon* **2012**, *50*, 3799.

Question 2: *As LiNO_3 is employed in the electrolyte, the high coulombic efficiency may not be directly attributed to the anchoring effect. Have you tried the case without LiNO_3 ?*

Response 2: Lithium nitrate is the most common additive for electrolytes used in Li-S systems, which passivates metallic lithium and has an influence on polysulfide shuttle suppression. In the pioneering work, Aurbach et al. proposed a working mechanism for the LiNO_3 additive, which is based on the formation of a lithium protective film in situ (*J. Electrochem. Soc.* **2009**, *156*, A694). Further studies confirmed a positive effect of LiNO_3 , including polysulfide-shuttle suppression and reduced contact resistance at the lithium electrode. A majority of the previously reported Li-S cells with ether-based electrolytes, contain lithium nitrate in their electrolyte composition. In our study, we also tested the electrochemical performances of the materials in the electrolyte without LiNO_3 additive according to the comments of the reviewers. The results showed that the Coulombic efficiency of the electrode material is slightly reduced in the electrolyte without LiNO_3 , which confirms that the electrolyte additive is important for Li-S batteries. Therefore, the contribution of LiNO_3 to the performance improvement should not be ignored. However, the electrochemical performance of VN/G composite electrode was better than that of RGO electrode in the same electrolyte without LiNO_3 . These results demonstrate the advantages of VN as the host material for lithium sulfur batteries.

In order to express more accurately, we have added the sentence “The LiNO₃ additive in the electrolyte also has a positive effect on the Coulomb efficiency and cyclic performance of Li-S batteries.” after the sentence “The VN/G cathode delivered an excellent initial discharge capacity of 1471 mAh g⁻¹ with a Coulombic efficiency above 99.5%, and more importantly, it was able to maintain a stable cycling performance for 100 charge-discharge cycles at 0.2 C, indicating that dissolution of polysulfides into the organic electrolyte was effectively mitigated in the VN/G electrode.” We have added the literature (*J. Electrochem. Soc.* **2009**, *156*, A694) in the revised manuscript as reference 36 and labeled in the manuscript.

Question 3: Please provide the theoretical conductivity information of VN in the revised text. What is the conductivity of the Graphene/VN composite?

Response 3: As shown in the table below (also see Supplementary Table 1), many metal nitrides have a high electrical conductivity comparable to their metal counterparts. The theoretical conductivity of VN is about 1.17×10^6 S m⁻¹, which is larger than that of reduced graphene oxide. The addition of VN can greatly improve the electrical conductivity of VN/G composites. Furthermore, the electrical conductivity of the VN/G composite is ≈ 1150 S m⁻¹ measured by the four-point probe method, which is over 4 times larger than that of reduced graphene oxide (about 240 S m⁻¹). Therefore, the electrons involved in the charging and discharging processes can transport very fast in the nested network of the VN/G cathode, which is favorable for fast polysulfide conversion.

Supplementary Table 1 Electrical conductivity of different metal nitrides at room temperature

Materials	VN	TiN	Mo ₂ N	WN	Ni ₃ N
Conductivity ($\times 10^6$ S m ⁻¹)	1.17	4.0	5.05	11.1	0.36

Reference: S. T. Oyama, Introduction to the chemistry of transition metal carbides and nitrides, In *The chemistry of transition metal carbides and nitrides*, (Ed: S. T. Oyama), Blackie Academic and Professional, **1996**.

We have given the theoretical electrical conductivity of VN, and also added the sentence “In addition, the VN/G composite also exhibits an electrical conductivity of $\approx 1150 \text{ S m}^{-1}$ measured by the four-point probe method, which is over 4 times larger than that of RGO (about 240 S m^{-1}).” after the sentence “The VN/G cathode has a smaller resistance ($28 \text{ } \Omega$) than that of the RGO cathode ($95 \text{ } \Omega$), which can be explained by the high electrical conductivity of metal nitrides comparable to their metal counterparts, as shown in Supplementary Table 1.” in the revised manuscript.

Question 4: *Figure 1a, the schematic for the GO should be revised to reflect the existence of oxygen containing groups.*

Response 4: According to the referee’s suggestion, we revised the schematic of the graphene oxide, as shown below. In the modified schematic, we used red and cyan ball to represent oxygen and hydrogen atoms, respectively, which reflects the existence of oxygen functional groups in the graphene oxide more clearly.

The schematic of graphene oxide

We also replaced the original schematic in Figure 1 with a new schematic of graphene

oxide in the revised manuscript.

Response to Reviewer 3

Overall comments: Sun et al. report the application of a VN/graphene composite as a host material for a catholyte Li-S cell. The highlighted scientific advance was the use of a conductive metal nitride material and its catalytic properties towards polysulfide redox. However, there are some concerns that prevent recommendation for publishing in Nature Commun.

We thank this referee for the efforts on evaluating our work.

Question 1: There have been previous reports on using conductive metal nitrides, TiN (Adv. Mater. 2016, DOI: 10.1002/adma.201601382, J. Mater. Chem. A, DOI: 10.1039/C6TA07411A), which shows the benefits of adding or using nitrides in sulfur host materials. This may compromise the significance of this report

Response 1: We carefully read the two papers (Adv. Mater. 2016, DOI: 10.1002/adma.201601382; J. Mater. Chem. A DOI: 10.1039/C6TA07411A) that just published. We found that pure TiN was used as a host material in both papers, and chemical adsorption and high conductivity of TiN for polysulfides are used to inhibit polysulfide shuttling and promote polysulfide conversion. And only the adsorption of polar polysulfides is considered, while the re-deposition of non-polar charging product sulfur is neglected. In our VN/Graphene model system, we use strong adsorption and high conductivity of VN to inhibit shuttle effect and to facilitate efficient conversion of polysulfides. At the same time, the nonpolarity of graphene in the VN/G composite is beneficial for the re-deposition of the charging product sulfur. The hetero-polar VN/G electrodes provide both polar (VN) and nonpolar (graphene) platforms to facilitate the binding of solid Li_xS and sulfur species to the electrode. In addition, we also found that VN can improve the kinetics of the redox reaction in a lithium sulfur battery because VN has similar catalytic activity to that of precious metals. However, TiN does not have such a catalytic effect. Therefore, our selected VN as a host material for lithium sulfur batteries has significant advantages and novelty, compared to TiN.

Question 2: There have been quite a few studies on using oxides. The authors have to demonstrate the true advances of using nitrides over oxides, or at least compare the performances with the oxides-based hosts, e.g VO_x, in order to highlight the significance of using VN.

Response 2: According to the referee's suggestion, we synthesized VO_x/G composites using a process similar to that for the synthesis of VN/G composite except that the atmosphere of heat treatment was changed from ammonia to argon, and the electrochemical performances of the VO_x/G cathode were tested. As can be seen from the figure below (Figure R1), although VO_x/G also has good cycling stability (the initial capacity was 951 mAh g⁻¹ and retained 67% of the initial capacity after 100 cycles), which is due to the chemical adsorption of VO_x for polysulfides, but the Coulomb efficiency (only 93%) is significantly reduced after 100 cycles. The low Coulombic efficiency probably resulted from the low converting rate of surface-bound sulfur species on VO_x, which led to surface poisoning by unreacted polysulfides, prevented the subsequent adsorption, and weakened its suppression of polysulfide shuttle. The results further show that the highly conductive VN can achieve adsorption and rapid conversion of polysulfides, thus lead to the high electrochemical performance of cathode for the lithium sulfur battery.

Figure R1 (a) Cycling stability at 1C and (b) rate performance of the VO_x/G cathode.

The Figure R1 was added in Supplementary Fig.9 in the revised manuscript. We have also added the sentence “In contrast, the VO_x/G electrode displayed rapid capacity decay and low Coulombic efficiency (about 93% after 100 cycles), which probably resulted from the low conversion efficiency of polysulfides adsorbed on non-conductive VO_x surfaces (Supplementary Fig.9).” before the sentence “The excellent electrochemical performance of the VN/G cathode can be attributed to the following factors.” in the revised manuscript.

Question 3: *The use of 70 wt% of porous graphene sponge requires a large amount of electrolyte, >30 ul/mg sulfur as the authors report. This lowers the volumetric energy density. Even still, there have been reports of using graphene based catholyte cells that apply 8.5 mg cm⁻² of sulfur loading or more (Nature Commun. 2015, 6, 7760). Good performance at higher loading have to be demonstrated in this manuscript.*

Response 3: Thank the referee for constructive suggestion. The density of the VN/G composite macroform is 276 mg cm^{-3} , which is more than two times higher than that of graphene macrostructure prepared by the same freeze-drying method ($<100 \text{ mg cm}^{-3}$ in most cases, *Energy Environ. Sci.*, 2015, 8, 1390 and *Sci. Rep.* 2013, 3, 2975). Controlling the drying process is a simple but effective way to simultaneously tailor the structures and properties of graphene-based macrostructure, such as pore structure, density, mechanical strength, conductivity etc. Therefore, the density of our VN / G composites can be adjusted according to different requirements. In addition, a large packing pressure (about 7MPa) during the assembly of a coin cell can compact the VN/G electrode, further increasing the density of the electrodes and reducing the amount of electrolyte used. Therefore, it is possible to further increase the volumetric energy density of the battery by increasing the density of the electrode.

We agree with the reviewer's comments that the cathode with high sulfur loading is very important for the application of lithium sulfur batteries. As pointed out by the reviewers, there are significant progresses on Li-S batteries with high sulfur loading very recently. For example, we reported a 3D hybrid graphene hierarchical macrostructure as both a current collector and a host for sulfur in Li-S batteries, and this material can achieve remarkably high sulfur loading of 14.36 mg cm^{-2} and sulfur content of 89.4 wt% simultaneously. (*Adv. Mater.* **2016**, 28, 1603-1609). In the article mentioned by the reviewer (*Nature Commun.* **2015**, 6, 7760), the authors used the N,S-codoped graphene with high specific surface area as a 3D scaffold to accommodate high active material loading. In the above studies, the high specific surface area and high electrical conductivity of the host materials are necessary for achieving a high sulfur loading cathode. Compared to the N,S-codoped graphene (its specific surface area is $171.4 \text{ m}^2 \text{ g}^{-1}$) reported in the literature, our VN/G composites have a small specific surface area ($37 \text{ m}^2 \text{ g}^{-1}$). The host materials with a low surface area are unable to adsorb a high amount of polysulfides, and therefore, it is very difficult to obtain a high content of sulfur. In our manuscript, we propose a new concept that uses the high conductivity and catalytic effect of metal nitrides to inhibit the shuttle effect and promote the kinetics of polysulfide conversion reactions. We

believe that if we can prepare metal nitrides with a high specific surface or their composite materials, we can achieve high sulfur loading electrodes and obtain excellent electrochemical performance.

In addition, we also note that the sulfur loading of some recently reported cathodes containing an inorganic polar host material is relatively low, but they proposed a new method to suppress the shuttle effect and improve the performance of sulfur cathode, and opened a new direction to fabricate high-performance advanced Li-S batteries. For example, the sulfur loading in a TiN based cathode mentioned by the reviewer was only 1.0 mg cm^{-2} (*Adv. Mater.* **2016**, DOI: 10.1002/adma.201601382), a MnO_2 based cathode with an average sulfur loading between 0.7 and 1.0 mg cm^{-2} have also reported (*Nature Commun.* **2015**, 6, 5682), and the sulfur loading in a Ti_4O_7 -based cathode was 1.5 - 1.8 mg per electrode (*Nature Commun.* **2016**, 7, 11203).

Question 4: *Some other details that can help improve the manuscript. The surface area of rGO needs to be reported and compared with VN-rGO.*

Response 4: According to the referee's suggestion, we evaluated the specific surface area of the RGO by nitrogen adsorption-desorption method. As shown in the figure below, the RGO has a high specific surface area of $296 \text{ m}^2/\text{g}$ and a hierarchical pore structure.

Figure R2 Nitrogen adsorption-desorption isotherm of the RGO. Inset: the pore size distribution obtained using the BJH method.

We included the nitrogen adsorption-desorption isotherm of the RGO (Figure R2) as Supplementary Fig. 4, and have added the sentence “In contrast, the specific surface area of the RGO was as high as $296 \text{ m}^2 \text{ g}^{-1}$ (Supplementary Fig. 4).” after the sentence “The specific surface area of the VN/G was $37 \text{ m}^2 \text{ g}^{-1}$ with mesopores 18 nm in diameter (Supplementary Fig. 3), which is consistent with the TEM observation.” and labeled in the revised manuscript.

Question 5: *In Figure S5, what is the condition of the cell for EIS? Pristine or in discharged/ charged states? This is important for claiming the affinity between VN and polysulfides.*

Response 5: We thank the reviewer very much for the valuable suggestion. Our electrochemical impedance spectra are derived from pristine cells before cycling and recorded from 10 kHz to 100 MHz at open circuit voltage at room temperature. In order to describe more accurately, we modified the original figure caption of the Figure S5 to “Comparison of the electrochemical impedance spectra of the VN/G and RGO cathodes before cycling. The data was recorded from 10 kHz to 100 MHz at open circuit voltage at room temperature.” in the revised manuscript.

Question 6: *In Figure 4d, the rate capability of rGO cell has to be compared directly.*

Response 3: For comparison, we have added the rate performance of the RGO electrode. As shown below, the RGO electrode shows a lower discharge capacity than the VN/G cathode. This result further indicates that the VNG electrode has better electrochemical performance.

Figure R3 Rate performance of the RGO cathode at different current densities.

We have revised the Figure 4d, and also added the sentence “In contrast, the RGO electrode exhibited lower discharge capacity and poorer stability under the same conditions.” after the sentence “As shown in Figure 4d, when the electrode was cycled at different rates of 0.2 C, 0.5 C, 1 C, 2 C and 3 C, the cell was able to deliver discharge capacities of 1447, 1241, 1131, 953, 701 mAh g⁻¹, respectively.” in the revised manuscript.

Question 7: In Figure 5, the binding energy of VN with Li₂S₆ has to be compared with N-doped graphene, instead of plain graphene, in order to correlate with the experiments.

Response 7: We agree with the referee that the binding energy of VN with Li₂S₆ should be compared with N-doped graphene, instead of pristine graphene, for a more precise correlation with our experiments. Actually, the binding energy between Li₂S₆ and N-doped graphene ranges from 0.7 to 2.85 eV for different N-doping configurations, as reported in our recent theoretical results (L. C. Yin et al., Nano Energy, 2016, 25, 203-210). Considering that the pyridinic-N is the dominant dopant in N-doped graphene synthesized in this work, as shown in the Supplementary Figure 6, we compare the binding energy between VN and Li₂S₆ with that between Li₂S₆ and pyridinic-N-doped graphene in the revised manuscript.

In order to make this point clearly, we changed the statements “For comparison, the

binding energy between Li_2S_6 and graphene was also considered, and was calculated to be 0.74 eV. In contrast, the binding energy between Li_2S_6 and VN was calculated to be much larger (3.75 eV). This is mainly due to the much stronger polar-polar interaction between Li_2S_6 and VN than the polar-nonpolar interaction between Li_2S_6 and graphene. In comparison with the case of Li_2S_6 on graphene (Fig. 5b), the strong polar-polar interaction between Li_2S_6 and VN results in an obvious deformation of the Li_2S_6 molecule (Fig. 5c), forming three S-V and one Li-N bonds.” to “As shown in the Supplementary Figure 7, the pyridinic-N is the dominant dopant in N-doped graphene synthesized in this work. For comparison, the binding energy between Li_2S_6 and pyridinic N-doped graphene was considered, and it has been reported to be 1.07 eV³⁸. In contrast, the binding energy between Li_2S_6 and VN was calculated to be much larger (3.75 eV). This is mainly due to the much stronger polar-polar interactions between Li_2S_6 and VN than those between Li_2S_6 and pyridinic N-doped graphene. In comparison with the case of Li_2S_6 on pyridinic N-doped graphene (Fig. 5b), the strong polar-polar interaction between Li_2S_6 and VN results in an obvious deformation of the Li_2S_6 molecule (Fig. 5c), forming three S-V and one Li-N bonds” in the revised manuscript. We also modified the Figure 5b by using N-doped graphene model instead of original graphene model.

Question 8: *On line 251, page 13, the bonding length in original literature needs to be provided for comparison.*

Response 8: According to the referee’s suggestion, we added the calculated bond lengths of V-S and Li-N for comparison in the revised manuscript. We modified the sentence “The bond lengths of these S-V and Li-N bonds are very close to those the corresponding bonds in bulk VS and LiNH_2 (2.42 Å and 2.06 Å)” to “The bond lengths of these S-V (2.49-2.61 Å) and Li-N (2.08 Å) bonds are very close to the corresponding bond lengths in bulk VS (2.42 Å) and LiNH_2 (2.06 Å)” in the revised manuscript.

Response to Reviewer 3

Overall comments: The authors have now properly addressed the questions and highlighted the major contribution of this paper to the Li-S community. It can be accepted at this stage.

We thank this referee for the very positive comments.